# Design and Verification of Piano Playing Assisted Hand Exoskeleton Robot

**DOI:** 10.3390/biomimetics9070385

**Published:** 2024-06-25

**Authors:** Qiujian Xu, Dan Yang, Meihui Li, Xiubo Ren, Xinran Yuan, Lijun Tang, Xiaoyu Wang, Siqi Liu, Miaomiao Yang, Yintong Liu, Mingyi Yang

**Affiliations:** 1School of Arts and Design, Yanshan University, Qinhuangdao 066004, China; xuqiujian@ysu.edu.cn (Q.X.); yuanxinran0112@163.com (X.Y.); liusiqi19990919@163.com (S.L.); yangmiaomiao07@163.com (M.Y.); 2YSU & DCU Joint Research Centre for the Arts, Music College, Daegu Catholic University, Daegu 38430, Republic of Korea; yangdan6413@163.com (D.Y.); 18686969513@163.com (M.L.); xiuboren@gmail.com (X.R.); wangxiaoyu_9607@163.com (X.W.); 13831481889@163.com (Y.L.); 3Department of Medical Assistant, Mount Eagle Univesity, Winston Salem, NC 27106, USA; lijuntang2007@gmail.com

**Keywords:** piano playing, finger typing, joint angles, hand exoskeleton, linkage mechanism design

## Abstract

Finger technique is a crucial aspect of piano learning, and hand exoskeleton mechanisms effectively assist novice piano players in maintaining correct finger technique consistently. Addressing current issues with exoskeleton robots, such as the inability to provide continuous correction of finger technique and their considerable weight, a novel hand exoskeleton robot has been developed to enhance finger technique through continuous correction and reduced weight. Initial data are gathered using finger joint angle sensors to analyze movements during piano playing, focusing on the trajectory and angular velocity of key strikes. This analysis informs the design of a 6-bar double-closed-loop mechanism with an end equivalent sliding pair, using analytical methods to establish the relationship between motor extension and input rod rotation. Simulation studies assess the exoskeleton’s motion space and dynamics, confirming its capability to meet structural and functional demands for accurate key striking. Prototype testing validates the exoskeleton’s ability to maintain correct finger positioning and mimic natural strike speeds, thus improving playing technique while ensuring comfort and safety.

## 1. Introduction

The piano, as an ancient and captivating instrument, has consistently captivated numerous music enthusiasts and artists [1]. However, beginners in the process of learning piano playing often encounter various challenges, with one of them being maintaining the correct finger technique [2,3]. Finger technique is a fundamental aspect of piano instruction and, at the same time, the most crucial [4]. Correct finger technique is crucial for piano performance; it not only contributes to musical expression and skill but also helps prevent hand injuries [5,6].

Typically, the teaching and correction of finger technique for piano beginners are guided by music teachers, incurring high labor costs, and it’s challenging for beginners to consistently maintain the correct finger technique. With the development of exoskeleton technology, hand-assist exoskeletons are increasingly applied in the field of piano learning and performance [7,8]. Cai Rongjie et al. utilized Leap Motion to establish a finger-touch action measurement platform, collecting parameters of finger-touch actions and obtaining the movement curve of finger-playing keys. They designed an exoskeleton piano teaching mechanical hand using a linkage mechanism and a cable transmission method for piano instruction and training [9,10]. Takahashi et al. developed a novel soft exoskeleton glove using pneumatic muscle technology, capable of producing dexterous finger joint movements. It includes 20 degrees of freedom, minimizing restrictions on free finger movement, and generating 8N of static force at the fingertips [11]. Lin et al. proposed a soft robot hand exoskeleton combined with machine learning algorithms. Each fingertip integrates a resistive sensor array and 16 pneumatic feedback units to assist users in experiencing finger sensations during piano playing [12]. DigituSync, a passive hand exoskeleton designed by Nishida et al., physically connects two hands through multiple four-bar linkages. This allows two users to transmit finger movements and pressing forces in real-time, facilitating music teachers in more efficiently and accurately assisting students in piano playing learning [13].

Despite the emergence of various rigid and flexible hand exoskeleton technologies for assisting piano beginners in the learning process, these technologies still face several limitations in practical applications: (1) Although flexible exoskeletons offer high comfort and adaptability to hand morphology, they fail to provide sufficient continuous support to ensure students consistently maintain the correct finger technique during playing. (2) Rigid exoskeletons have a large volume and weight, resulting in low comfort and exerting additional pressure on the wrist. (3) When exoskeletons regulate and correct finger movements, they pay less attention to joint rotational angular velocity, affecting the natural speed of finger tapping movements and reducing the exoskeleton’s effectiveness in terms of standardization, comfort, and safety.

Addressing the aforementioned issues, this paper proposes a fully-driven, lightweight piano playing assistance exoskeleton robot for the hands. Based on the characteristics of correct finger striking movements, a fully-driven linkage configuration is designed to continuously and effectively assist performers in maintaining the correct finger technique, preventing changes in finger technique due to incorrect habits and muscle fatigue. Utilizing a dual-closed-loop linkage transmission structure and manufactured through high-strength lightweight plastic 3D printing, the exoskeleton has a lightweight overall design with a small volume, providing high comfort and minimal pressure on the hands and wrists. The exoskeleton actuator can simulate the natural joint angular velocity of finger tapping. While standardizing tapping movements, it also maintains the natural angular velocity of the fingers, enhancing the standardization, comfort, and safety of the actions.

The content of this paper unfolds as follows.

Analyzing the physiological structure of the fingers and the action of striking piano keys to determine the auxiliary movement form of the exoskeleton structure.Designing an exoskeleton structure that meets the movement requirements based on the characteristics of finger striking piano keys.Conducting simulation analysis of the motion space and transient dynamics of the exoskeleton robot.Constructing a prototype of the exoskeleton robot and conducting experimental validation.

## 2. Methods

### 2.1. Analysis of Finger Tapping on Piano Keys

#### 2.1.1. Analysis of Basic Playing Finger Patterns

In piano playing, different performers develop unique finger technique characteristics. However, for piano beginners, it is essential to learn to play according to scientific and standard finger techniques.

The above process consists of 4 steps: Begin by lightly placing both hands in a fist on the white keys (As shown in subfigure ① of Figure 1); Next, relax the fists and keep the distal segment pointing downward perpendicular to the piano keys’ surface (except for the thumb) (As shown in subfigure ② of Figure 1). Then, support the metacarpophalangeal (MCP) joints; Additionally, maintain the proximal interphalangeal (PIP) joints and distal interphalangeal (DIP) joints in a protruding state (As shown in subfigure ③ of Figure 1); Finally, gently lift the palms, forming a rounded arch with the entire hand, ensuring that the wrist joints are parallel to the keyboard (As shown in subfigure ④ of Figure 1). Throughout the entire process, there are two key points regarding the finger movements: (1) The three joints of the fingers (except for the thumb) should remain prominent, avoiding any joint collapse; (2) After the fingers touch the keys, the distal segment should always remain parallel to the surface of the piano keys.

#### 2.1.2. Analysis of the Motion of Knocking Qin Keys

##### Sports Data Collection Methods

To accurately capture the motion trajectory and parameters when fingers correctly strike piano keys, several sensing devices can be employed, including 3D motion sensing devices based on video analysis [14,15,16,17,18], inertial sensing devices based on nine-axis gyroscopes [19], and image sensing devices based on optical markers [20]. However, 3D motion sensing devices and optical marker measurement systems are expensive and require stringent conditions. They also exhibit significant measurement errors when applied to small joints such as fingers. Nine-axis gyroscopes, such as the MPU6050 sensor, measure the pose of a rotating object based on gravitational acceleration and calculate its pitch, yaw, and roll angles. However, when measuring single-dimensional angles, they are susceptible to interference from changes in posture in other dimensions, leading to significant errors in joint angle measurements, particularly in finger joints.

##### Finger Joint Angle Collection Experiment

As, in piano playing learning, the overall height of the hand is generally fixed, and the core of finger striking on piano keys lies in the rotation of finger joints. Therefore, this paper employs a self-made combined angle sensor to measure the changes in finger joint angles during piano key strikes, as shown in Figure 2a. The calibration method for the angle sensor is as follows: When the finger is fully extended, the angles of each joint are set to 0. As the joint bends, the joint angle becomes a positive value, and when overextended, the joint angle becomes a negative value, as illustrated in Figure 2b. After obtaining the rotation angle data of each joint, coupled with the physiological dimensions of the fingers, it is possible to calculate the motion trajectory and parameters of the end of each joint of the fingers.

The participants in the angle collection experiment were piano beginners who had received finger technique instruction, totaling 10 individuals, including 2 females and 8 males. The participants’ ages ranged from 20 to 25 years, with weights between 55 and 75 kg, heights between 170 and 179 cm. All participants were right-handed, in good health, and had no relevant hand-related medical history. The finger selected for data collection was the index finger of the right hand. The angle data collected in the experiment for the three joints are shown in Figure 3. The rising edge corresponds to the striking process, with joint angles gradually increasing, while the falling edge corresponds to the lifting process, with joint angles gradually decreasing.

According to the calibration method in Figure 2b, with the back of the hand and palm bones as the reference surface, the angle between the distal segment and the reference surface should be the sum of the three joint angles, with an average of 89.5267°, as shown in Figure 4.

##### Analysis of the Motion Trajectory of Knocking Qin Keys

Based on the physiological measurement data of the finger, the lengths selected are 50 mm for the proximal segment (from MCP joint to PIP joint), 25 mm for the middle segment (from PIP joint to DIP joint), and 27 mm for the distal segment (from DIP joint to fingertip). Combining the mean angle data of each joint during multiple strikes, the motion trajectory of the fingertip during the striking process can be calculated, as shown in Figure 5. It can be visually observed that the distal segment remains approximately perpendicular to the horizontal plane throughout the striking process, with significantly smaller amplitude variations compared to the proximal and middle segments.

##### Finger Joint Angle Collection Experiment

Based on the analysis of the angle data in Figure 2a,b, the motion patterns during the striking and lifting processes in the playing activity are consistent, with opposite directions of motion. Therefore, only the striking process in the complete playing activity is extracted for analysis. The average duration of a single striking process is approximately 0.35 s. By capturing the angle change curves of each joint during each strike, equations for the temporal sequence of rotation angles for each joint can be fitted:(1)$$\alpha (t)=40.61t^{4}-69.15t^{3}+24.36t^{2}+0.851t-0.88$$
(2)$$\beta (t)=403.1t^{5}-489.2t^{4}+210.9t^{3}-33.9t^{2}-0.15t-0.34$$
(3)$$\gamma (t)=23.52t^{5}-157.5t^{4}+123.2t^{3}-29.99t^{2}+0.62t-0.46$$
where α(t) represents the MCP joint angle, β(t) represents the PIP joint angle, and γ(t) represents the DIP joint angle.

Taking the derivative of the curve equations for the striking motion allows obtaining their angular velocities. The angle and angular velocity curves for each joint are illustrated in Figure 6.

### 2.2. Exoskeleton Mechanism Design

#### 2.2.1. Configuration Selection

Based on the physiological structure of the four fingers excluding the thumb, a preliminary form of the exoskeleton mechanism can be established. The final exoskeleton configuration should include motion chains *l*_1_, *l*_2_, *l*_3_, parallel to the proximal, middle, and distal segments of the fingers, respectively. Additionally, the frame should be fixed on the metacarpal bone, as shown in the schematic diagram in Figure 7.

According to the finger technique and striking action requirements in piano playing, a crucial aspect is to keep the distal finger joint perpendicular to the piano keys throughout the entire process of lifting, dropping, and striking the finger. While sliding pairs can meet this action requirement, a mechanism that includes both sliding pairs and linkages will increase the overall volume and weight, which contradicts the lightweight design requirement. Therefore, a linkage mechanism needs to be used instead of sliding pairs. Typically, a four-bar loop can be employed to achieve approximate translational sliding, such as a parallel four-bar mechanism as shown in Figure 7. If a parallel four-bar mechanism is used instead of sliding pairs, the frame on the metacarpal bone needs to be directly connected to *l*_3_, as shown in Figure 8. In this case, the connecting rod DC crosses over *l*_1_ and *l*_2_, preventing the PIP joint from rotating, which does not meet the finger movement requirements for piano playing. To allow *l*_3_ to remain approximately perpendicular to the horizontal plane while allowing the PIP joint to rotate, this paper proposes a 6-bar double-closed-loop mechanism with an equivalent sliding pair at the end, as shown in Figure 8, The rod EF is parallel to *l*_3_, and the rods DG and GF are parallel to *l*_1_ and *l*_2_, respectively.

#### 2.2.2. Mechanism Design

In the 6-bar double-closed-loop mechanism with an equivalent sliding pair at the end, the movement of the connecting rod EF always remains parallel to the frame AD. This requires that the trajectories of points E and F are the same and remain parallel. Through the analysis of the motion trajectory of the distal segment in piano key striking in Section Analysis of the Motion Trajectory of Knocking Qin Keys, the trajectories of points E and F can be determined. Therefore, the linkage mechanism can be designed based on the predetermined motion trajectory [21].

Firstly, design the ABCD loop based on the predetermined motion trajectory of point E. As shown in Figure 9, establish a Cartesian coordinate system. Let the coordinates of point E be, then, from the left double-bar group shown in Figure 9, it can be derived:(4)$$\{\begin{array}{c} x=x_{A}+a\cos\theta_{1}+e\cos\theta_{2}-f\sin\theta_{2}\\ y=y_{A}+a\sin\theta_{1}+e\sin\theta_{2}+f\cos\theta_{2} \end{array}$$

By simultaneously solving the two equations in the system, the variable can be eliminated, resulting in:(5)$$\begin{array}{l} {\left(x-x_{A}\right)}^{2}+{\left(y-y_{A}\right)}^{2}+e^{2}+f^{2}-2[e(x-x_{A})+\\ f(y-y_{A})]\cos\theta_{2}+2[f(x-x_{A})-e(y-y_{A})]\sin\theta_{2}=a^{2} \end{array}$$

Similarly, applying the same calculations to the right double-bar group yields:(6)$$\begin{array}{l} {\left(x-x_{D}\right)}^{2}+{\left(y-y_{D}\right)}^{2}+g^{2}+f^{2}-2[f(y-y_{D})-\\ g(x-x_{D})]\cos\theta_{2}+2[f(x-x_{D})+g(y-y_{D})]\sin\theta_{2}=c^{2} \end{array}$$

Equations (5) and (6) both contain the generalized coordinate, and they can be solved simultaneously, involving $x_{A},y_{A},x_{D},y_{D},a,c,e,f,g$ nine parameters. Therefore, it is possible to achieve precise design by using 9 predetermined points in the trajectory of point E. However, in practical applications, since Equations (5) and (6) form a second-order nonlinear system, their solution relies on numerical methods. As the number of predetermined points increases, the number of equations grows exponentially, making the solution increasingly difficult. Moreover, these systems of equations often have no solution in typical cases. Even if a solution exists, it may lose practicality due to the inability to meet practical requirements such as rod length ratios and transmission angles. To facilitate the solvability of the system of equations, and considering the physiological dimensions of the finger, parameters $x_{A}=0$, $y_{A}=0$, $x_{D}=30$, $y_{D}=0$, and f=0 are given. With these parameters fixed, there are only four remaining parameters a,c,e,g, and precise design can be achieved by using 4 predetermined points in the trajectory of point E.

After determining the linkage loop ABCD, further design the loop ABEFGD based on the predetermined trajectory of point F. Since the loop ABEFGD shares the linkages AB, BE, and GD with the loop ABCD, as shown in Figure 10. The right-side linkage group FGD of the loop, as shown in Figure 10, according to the given condition that EF is moderately parallel to AD, the projection of the right-side linkage group FGD on the *y*-axis is equal to that of the left-side linkage group ABE, leading to:(7)$$a\sin\theta_{1}+e\sin\theta_{2}=j\sin\theta_{3}+i\sin\theta_{4}$$

In which, parameters a and e, generalized coordinates $\theta_{1}$, $\theta_{2}$ and $\theta_{3}$ are all determined by the loop ABCD. Only the parameters i and j are unknown. Precise solution can be obtained by using 2 predetermined points in the trajectory of point F.

#### 2.2.3. Exoskeleton Configuration Design

After solving the length parameters of the 6-bar double-closed-loop mechanism with end equivalent sliding pairs, the exoskeleton structure can be initially assembled. The exoskeleton is designed to be driven by a linear motor, with link AB serving as the input link. It is necessary to adjust the size of the input link to ensure that the motor stroke and speed can meet the bending angle and angular velocity of the finger in Section 2.2.2. First, establish the geometric relationship between the input link and the linear motor, as shown in Figure 11.

In Figure 11, d represents the length of the input link based on its length at the rear of the frame, l represents the length of the motor, Δl represents the motor extension, and Δθ represents the rotation angle of the input link. Through the geometric relationship diagram of the linear motor, the mathematical relationship between the motor extension Δl and the rotation angle Δθ of the input link can be derived:(8)$$\left\{ \begin{array}{l} {\left(l+\Delta l\right)}^{2}={\left(l+d\right)}^{2}+d^{2}-2(l+d)d\cos\Delta \theta \\ \Delta \theta =arc\frac{{\left(l+d\right)}^{2}+d^{2}-{\left(l+\Delta l\right)}^{2}}{2(l+d)d} \end{array}\right.$$

By analyzing the mathematical relationship between the motor extension Δl and the rotation angle Δθ of the input link, it is possible to further calculate the extension and contraction speed ν of the motor based on the angular velocity of the index finger joint. This allows for the control of the exoskeleton’s speed:(9)$$\omega =\frac{d\theta }{dt}=\frac{d(\arccos\frac{{\left(l+d\right)}^{2}+d^{2}-{\left(l+\Delta l\right)}^{2}}{2(l+d)d})}{dt}$$

In the solving process, by substituting Δl=vt, the mathematical relationship between ν and ω can be established, which can be applied to the control algorithm of the exoskeleton motor:(10)$$\omega =\frac{2(\Delta l+1)v}{(1+d)d\sqrt{-\frac{{\left({\left(1+d\right)}^{2}+d^{2}-{\left(\Delta l+1\right)}^{2}\right)}^{2}}{{\left(1+d\right)}^{2}}+4}}$$

On the other hand, in order to make the exoskeleton structure more wearable, on the basis of the original exoskeleton structure, as shown in Figure 12a, finger clips were added to the connecting rods corresponding to the fingers for connecting the fingers, and the back of hand bracket located at the back of the hand was designed as a rack for the exoskeleton, and the exoskeleton was driven by a linear motor, which was mounted on a motor bracket, and the motor bracket was fixed to the forearm of the user, as shown in Figure 12b shown.

## 3. Results

### 3.1. Analysis of Exoskeleton Mechanism

#### 3.1.1. Workspace of the Exoskeleton Mechanism

The piano-assist exoskeleton mechanism consists of 6 components (including 1 frame), 2 closed-loop chains. According to the formula for degrees of freedom calculation:(11)F=N−2L
where N is the number of movable components, which is 5, and L is the number of closed loops, which is 2. Therefore, the degree of freedom of the exoskeleton mechanism is 1. When 1 driver is set up as shown in Figure 12b, the exoskeleton mechanism is fully actuated, and its workspace consists of deterministic motion trajectories.

Based on the joint angle analysis of the index finger during the tapping motion in Section Finger Joint Angle Collection Experiment, the rotation angle range of each joint can be obtained. According to the range of MCP joint angle, the motion range of the exoskeleton mechanism’s input rod can be estimated to be 0–47°. The theoretical motion trajectory of the exoskeleton compared to the finger’s motion trajectory is shown in Figure 13. The theoretical workspace of the exoskeleton can meet the requirements of finger tapping motion. It is expected that after wearing, it will significantly constrain the distal phalanx, assisting beginners in playing with a standardized finger posture and maintaining the correct finger posture.

#### 3.1.2. Transient Dynamic Finite Element Analysis

The exoskeleton mechanism undergoes complex forces during the striking motion, requiring consideration of the effect of inertia forces, resulting in nonlinear stress-strain changes. Therefore, transient dynamic analysis is necessary to validate the mechanical rationality and reliability of the mechanism.

Firstly, the 3D model of the piano-playing exoskeleton mechanism was created using Solidworks software (SolidWorks 2023 SP5.0) and saved in XT format. It was then imported into Ansys Workbench for simulation analysis. To reduce simulation computation time, a reasonable simplification was applied by removing the handback support and motor support, retaining only the linkage mechanism as the subject of the study.

In building the finite element model, considering the actual material selection for prototype construction, the linkage mechanism is made of a composite material, PLA with carbon fibers (PLA-K6CF, Kexcelled^®^—China, Suzhou, KEXCELLED Corporation, K6™ Series). In ANSYS, the material is defined with an elastic modulus of 6 × 10^3^ MPa, a Poisson’s ratio of 0.41, and a density of 1.04 g/cm^3^. Tetrahedral meshing was applied to the entire model, and the meshed model is shown in Figure 14.

After completing the model partition, it is necessary to set the boundary conditions. The boundary conditions are mainly divided into two categories: one is the constraint boundary conditions that restrict the relative motion of parts, and the other is the working boundary conditions of the mechanism. The motion constraint boundary conditions mainly apply rotational connections between the linkages, and fixed constraints are applied to the drive seat linkage. The working boundary conditions are based on the resistance of the piano keys, and the load is applied to the selected mechanism face with a force of 10 N, always along the positive *Y*-axis to simulate the force when striking the piano key. The input linkage is then subjected to a rotational angle of 47°.

By applying the boundary conditions of the linkage mechanism and then solving the calculations, the simulation results for equivalent stress are shown in Figure 15a,b, and the simulation results for equivalent strain are shown in Figure 16a,b.

According to the analysis of simulation results, the maximum stress is 11.9 MPa, and the maximum strain is 0.007 mm/mm, both occurring in the midsection of the input rod, parallel to the proximal phalanx. The simulation results are shown in Table 1, indicating that the exoskeleton linkage mechanism has sufficient mechanical strength.

## 4. Discussion

### 4.1. Prototype Construction and Experimental Verification

#### 4.1.1. Prototype Construction

The linkage and brackets of the exoskeleton robot prototype are made of a carbon fiber-embedded PLA material through 3D printing. The motor used is the L12 miniature linear actuator from the Canadian company Actuonix (Victoria, BC, Canada), with a stroke of 50 mm, a maximum thrust of 42 N, a maximum speed of 50 mm/s, and features speed and stroke adjustment functions. The controller’s main chip is the STM32F407, and it uses the TPS61088 boost chip to generate a 12 V voltage for motor operation. It can transmit driver stroke and speed information through Bluetooth communication and reserves 5 motor interfaces for future development of a full-hand exoskeleton. The overall structure of the exoskeleton prototype is shown in Figure 17.

#### 4.1.2. Workplace Validation

In order to verify whether the workspace of the exoskeleton meets the corrective requirements of piano playing, the performance was evaluated by collecting the angle changes of the three joints of the index finger after wearing the exoskeleton. The angle data collection in the experiment used the combination angle sensor mentioned in Section Finger Joint Angle Collection Experiment, which was combined with the exoskeleton for synchronized angle data collection without interfering with the motion of the exoskeleton, as shown in Figure 18.

The experimental subjects were selected as described in Section Finger Joint Angle Collection Experiment in the experiment to verify the workspace of the exoskeleton, first, the exoskeleton device was correctly worn on the index finger of the subjects. Ensure that the exoskeleton coordinates well with the hand structure and can be securely fixed on the finger. Then, the subjects sat in front of the piano in the correct posture, placed the hand wearing the exoskeleton at the appropriate height, and performed piano key strikes under the exoskeleton drive, as shown in Figure 18.

From Figure 19, it can be seen that after wearing the exoskeleton, the stability of the distal segment of the index finger during the striking motion is significantly improved, and the trajectory of the fingertip is concentrated near the theoretical trajectory of the exoskeleton end. After data analysis, the root mean square (RMS) values before and after wearing the exoskeleton are shown in Table 2, proving that the fluctuation of the fingertip positions of the index finger joints is significantly reduced after wearing the exoskeleton, and the stability is enhanced, which can continuously standardize and correct the striking motion of the index finger.

#### 4.1.3. Joint Angle and Angular Velocity Analysis

In Section 2.2.3, the mathematical relationship between the motor speed and the rotation angle of the input rod has been derived and applied to the motor control algorithm. In the experiment, by using the configuration in Figure 18a, the angles of each joint of the index finger were collected using angle sensors. The equation of the temporal sequence of angles was fitted, and the angular velocity curve was obtained through differentiation. This was then compared with the angles and angular velocities during the natural tapping state of the finger, as shown in Figure 13.

According to Figure 20, it can be seen that although the exoskeleton corrects the trajectory of the index finger after wearing, it does not significantly affect the natural joint mobility and tapping speed of the index finger. Through data analysis, the root mean square error (RMSE) of joint mobility and tapping speed of the index finger before and after wearing the exoskeleton is shown in Table 3, proving that the exoskeleton system has strong compliance with the natural angles and angular velocities during index finger tapping, meeting the requirements of standardization, comfort, and safety.

## 5. Conclusions

(1)By analyzing finger tapping actions and based on the fundamental finger techniques in piano playing, the characteristics of finger playing movements were determined for all fingers except the thumb. Further, using joint angle sensors, data on the joint angles of the fingers during playing movements were collected. A quantitative analysis of the characteristics of playing movements was conducted, obtaining the trajectory and angular velocity of striking the piano keys, and clarifying the design requirements for the exoskeleton to maintain vertical striking in the distal phalanx.(2)Based on the requirements of the exoskeleton design, a 6-bar double-closed-loop mechanism with an equivalent end sliding pair was selected. The design of the linkage mechanism was carried out using an analytical approach. Then, the mathematical relationship between the linear motor extension and the rotation angle of the input rod was established through geometric analysis. Combined with the physiological dimensions of the finger and wearable requirements, the design of the exoskeleton structure was completed.(3)After establishing the 3D model of the exoskeleton mechanism, simulation analyses were conducted on its workspace and transient dynamics. The results indicated that the theoretical workspace and structural strength of the exoskeleton could meet the requirements of the striking action. The construction of the experimental prototype further verified that the exoskeleton could enhance the stability of finger striking movements, simulate the natural speed of finger tapping, and, while regulating and correcting playing movements, ensure comfort and safety. This demonstrated the rationality and effectiveness of the exoskeleton structure design.

## Figures and Tables

**Figure 1 biomimetics-09-00385-f001:**
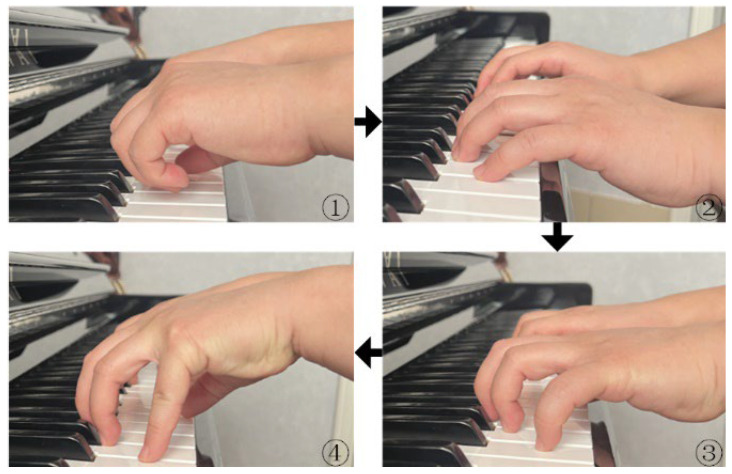
Standard finger playing process.

**Figure 2 biomimetics-09-00385-f002:**
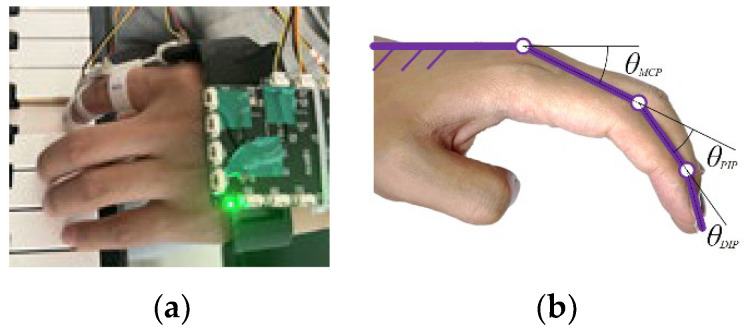
Combination angle sensor (**a**); Joint angle calibration (**b**).

**Figure 3 biomimetics-09-00385-f003:**
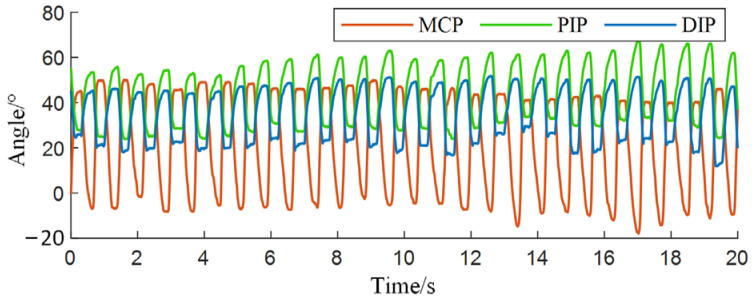
Angle data of each joint.

**Figure 4 biomimetics-09-00385-f004:**
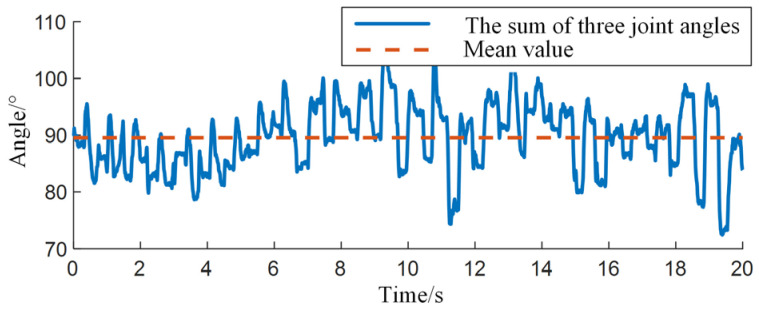
Analysis of Three Joint Angle Data.

**Figure 5 biomimetics-09-00385-f005:**
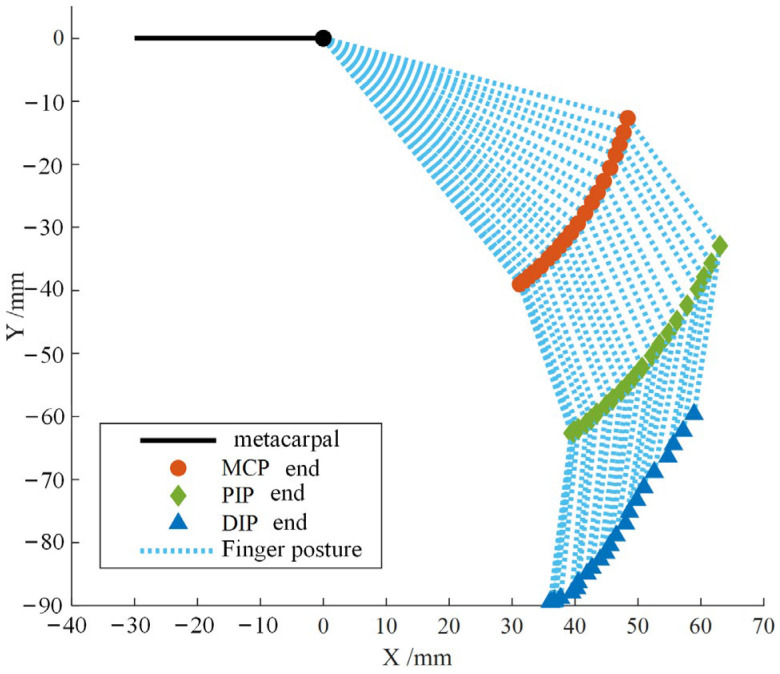
Trajectory of tapping motion at the end of each joint of the index finger.

**Figure 6 biomimetics-09-00385-f006:**
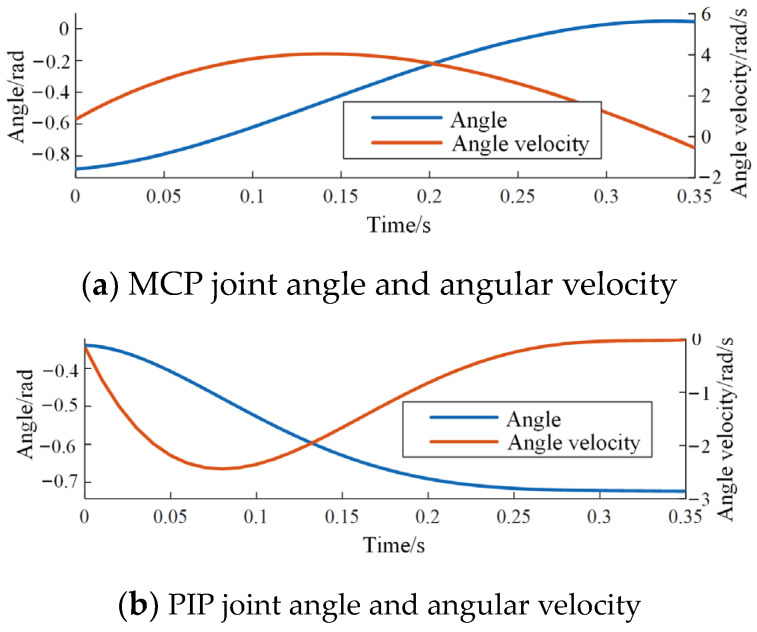
Angle and angular velocity of each joint in index finger tapping movement.

**Figure 7 biomimetics-09-00385-f007:**
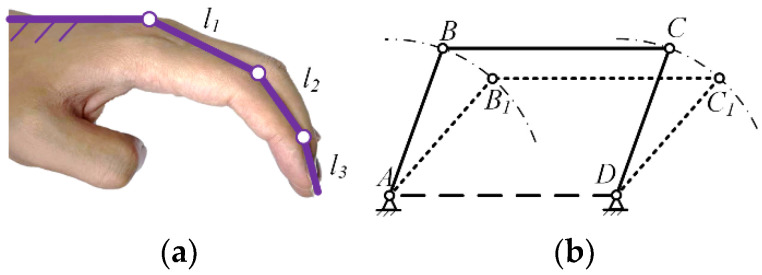
Index finger exoskeleton motion chain (**a**); Parallel four bar mechanism (**b**).

**Figure 8 biomimetics-09-00385-f008:**
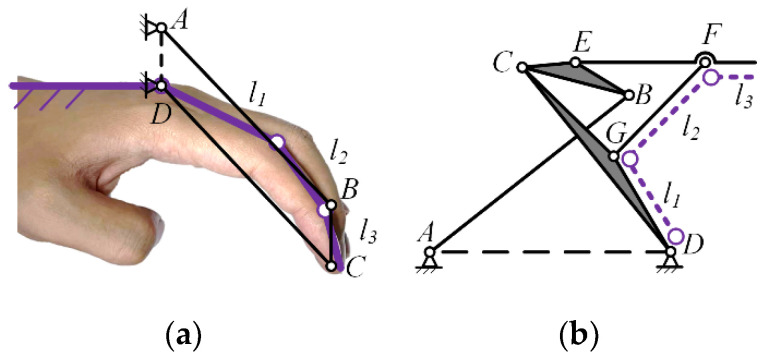
Parallel four bar mechanism replaces sliding pair (**a**); 6-bar mechanism (**b**).

**Figure 9 biomimetics-09-00385-f009:**
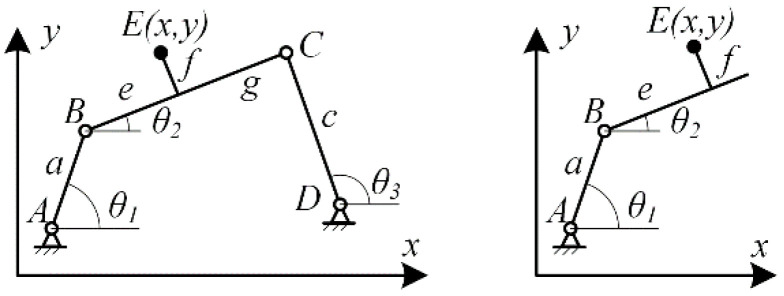
Loop ABCD analysis.

**Figure 10 biomimetics-09-00385-f010:**
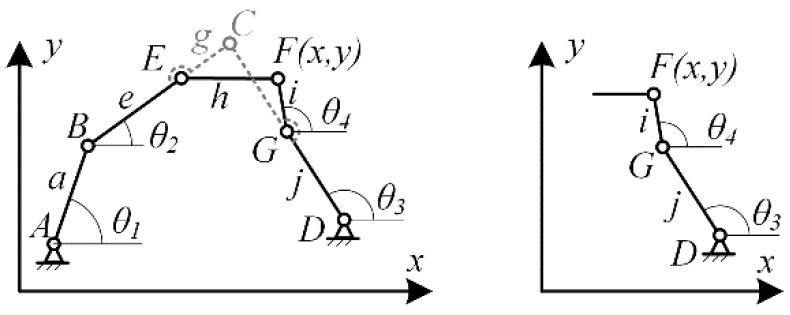
Analysis of the loop ABEFGD.

**Figure 11 biomimetics-09-00385-f011:**
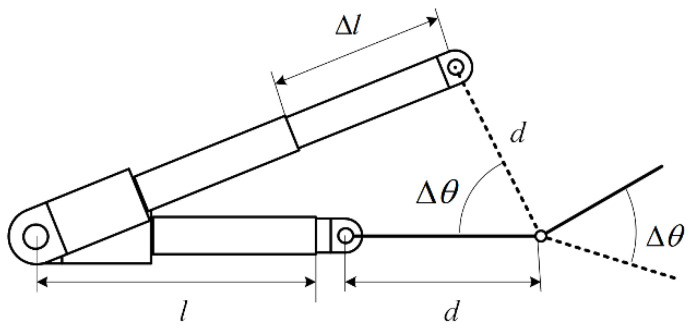
Geometric Analysis of Motor Extension and Input Link Rotation Angle.

**Figure 12 biomimetics-09-00385-f012:**
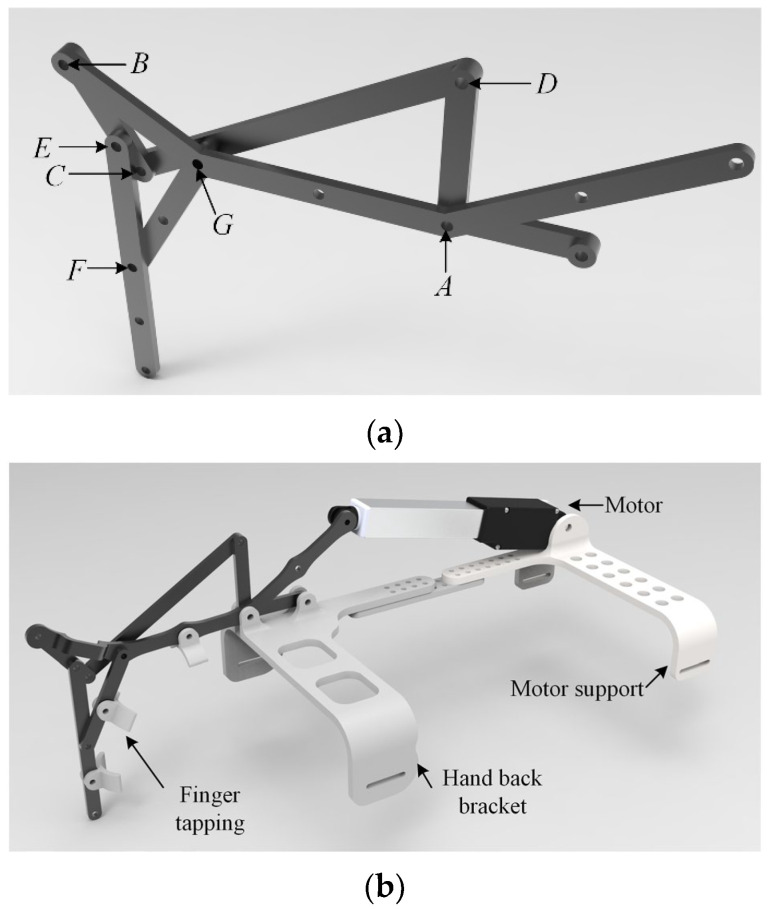
Exoskeleton mechanism (**a**); Exoskeleton robot model (**b**).

**Figure 13 biomimetics-09-00385-f013:**
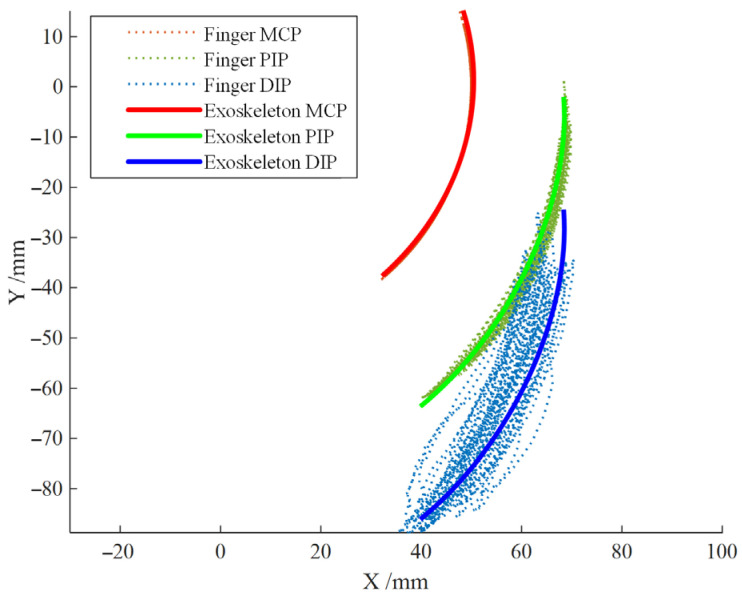
Comparison between Theoretical Trajectory of the Exoskeleton and Index Finger Trajectory.

**Figure 14 biomimetics-09-00385-f014:**
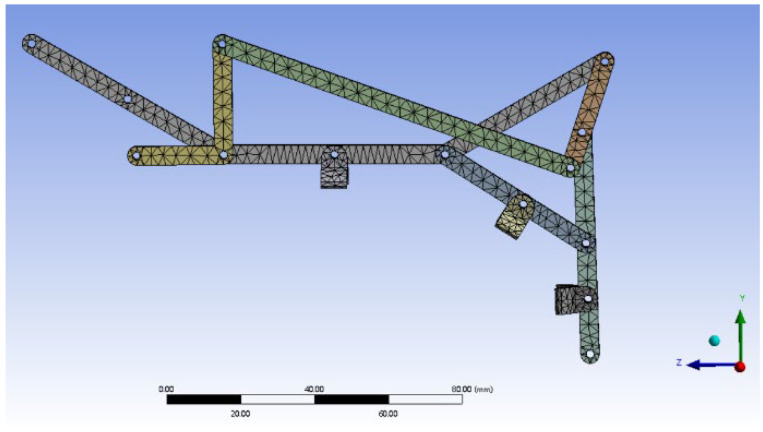
Model after meshing.

**Figure 15 biomimetics-09-00385-f015:**
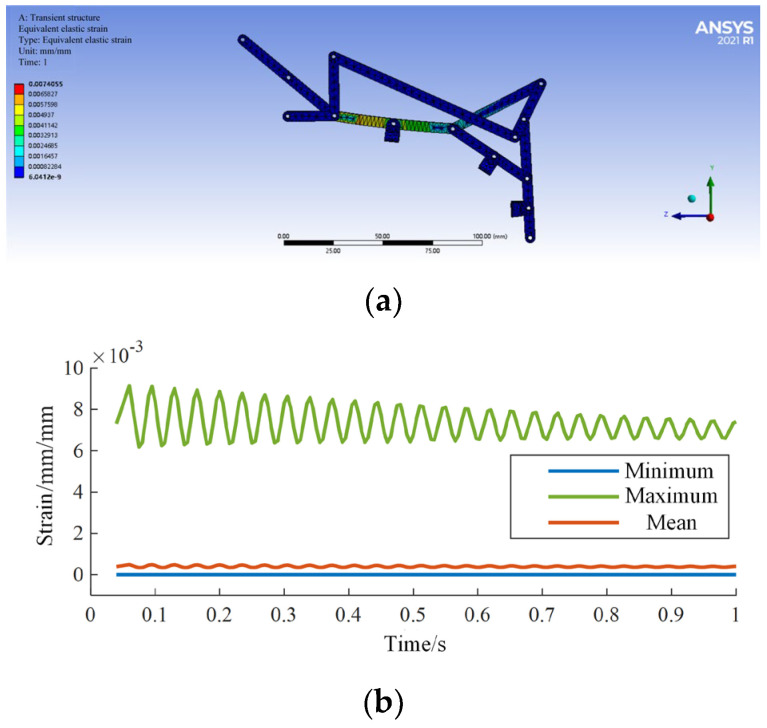
The equivalent strain simulation (**a**); Simulation results (**b**).

**Figure 16 biomimetics-09-00385-f016:**
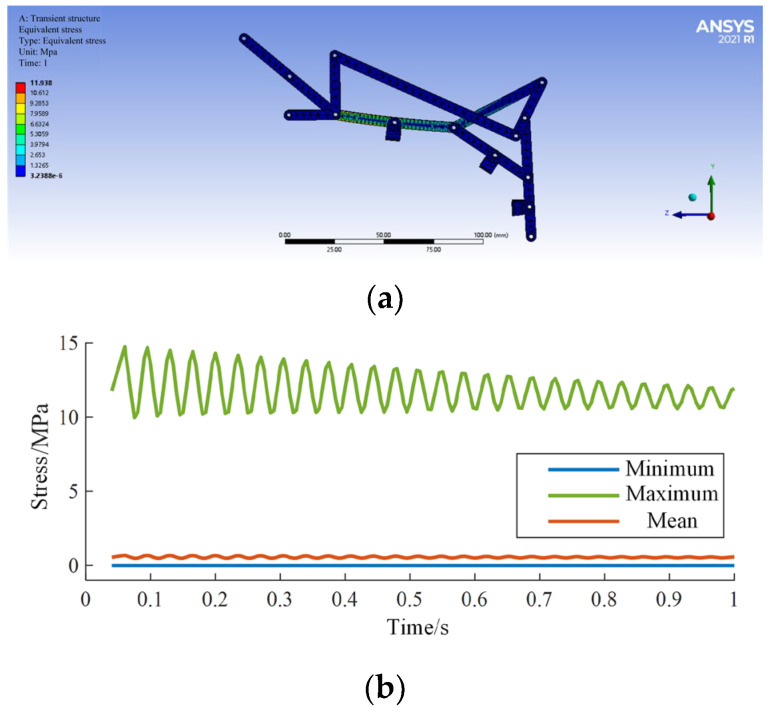
Equivalent stress simulation (**a**); Simulation results (**b**).

**Figure 17 biomimetics-09-00385-f017:**
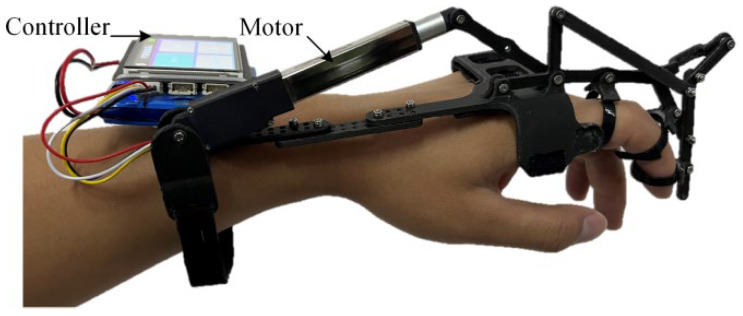
Exoskeleton robot prototype.

**Figure 18 biomimetics-09-00385-f018:**
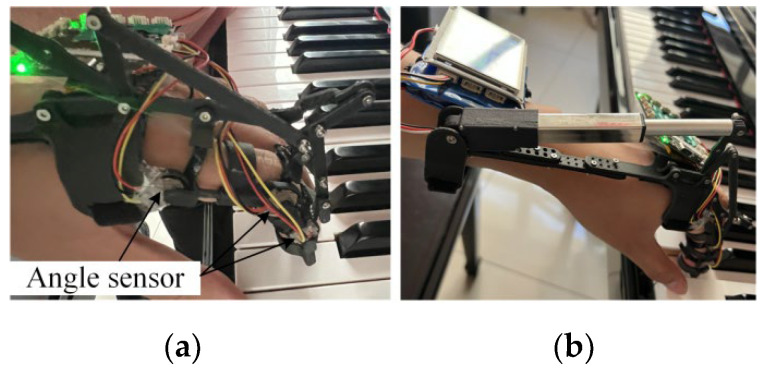
Exoskeleton mechanism combined with angle sensor (**a**); Exoskeleton robot driving the index finger to strike the piano key (**b**).

**Figure 19 biomimetics-09-00385-f019:**
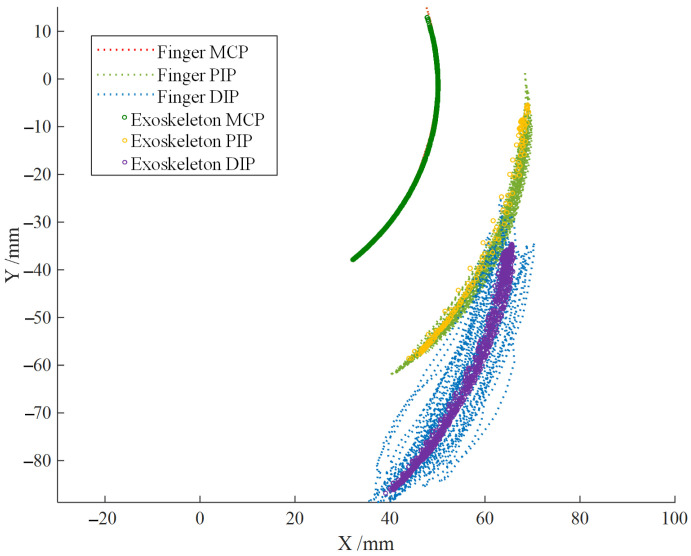
End trajectory of index finger joint before and after wearing exoskeleton robot.

**Figure 20 biomimetics-09-00385-f020:**
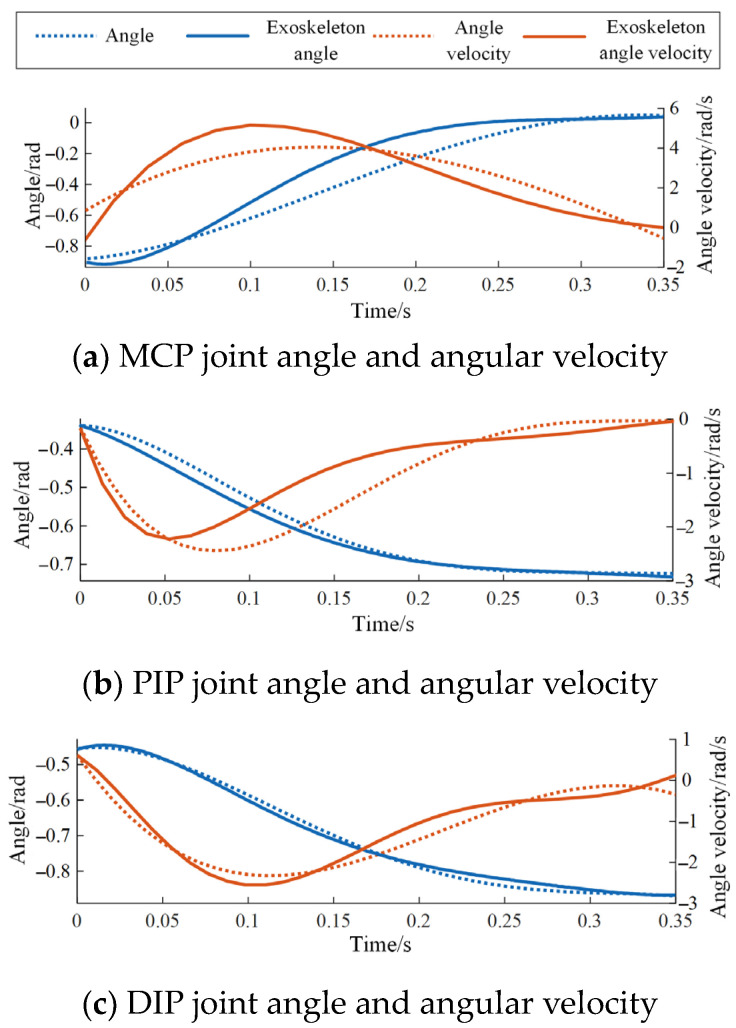
Joint angles and angular velocities of the index finger during piano playing before and after wearing the exoskeleton.

**Table 1 biomimetics-09-00385-t001:** Simulation result.

	Maximum Stress/MPa	Maximum Strain/mm/mm	Maximum Allowable Stress/MPa	Maximum Allowable Strain/mm/mm
Exoskeleton linkage	11.9	0.007	24.5	0.025

**Table 2 biomimetics-09-00385-t002:** RMS of the joint ends of the index finger before and after wearing the exoskeleton.

	MCP Joint End	PIP Joint End	DIP Joint End
Before wearing	0.5207	0.6413	0.7981
After wearing	0.4183	0.5201	0.5361

**Table 3 biomimetics-09-00385-t003:** Root Mean Square Error (RMSE) of Joint Angles and Angular Velocities of the Index Finger Before and After Wearing the Exoskeleton.

	MCP Joint	PIP Joint	DIP Joint
Joint Angles	0.2011	0.1537	0.1085
Joint Angular Velocity	0.2335	0.2367	0.1902

## Data Availability

The original contributions presented in the study are included in the article, further inquiries can be directed to the corresponding author/s.

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
