# Peer review of "Design and Verification of Piano Playing Assisted Hand Exoskeleton Robot"

_biomimetics, 2024, doi:10.3390/biomimetics9070385_

Round 1

Reviewer 1 Report

Comments and Suggestions for Authors

The manuscript discusses the design and verification of a hand exoskeleton robot to assist novice piano players in maintaining correct finger technique. Addressing the limitations of existing exoskeletons, such as lack of continuous support and excessive weight, the authors proposed a novel design featuring a 6-bar double-closed-loop mechanism with an end equivalent sliding pair. Finger movements during piano playing were analyzed using joint angle sensors, capturing data on trajectory and angular velocity. This tested if the exoskeleton's design enhanced finger technique through continuous correction while ensuring comfort and safety. Constructed from high-strength, lightweight plastic and driven by a linear motor, the exoskeleton's motion space and dynamics were validated through simulation studies, confirming its capability to meet structural and functional demands for accurate key striking. Prototype testing demonstrated the exoskeleton's effectiveness in maintaining correct finger positioning and mimicking natural strike speeds, thereby improving playing technique. Finite element analysis further indicated the exoskeleton has sufficient mechanical strength to withstand the forces encountered during piano playing. The proposed exoskeleton is therefore promising in supporting piano beginners by maintaining correct finger technique, reducing the likelihood of developing incorrect playing habits and muscle fatigue.

I found this study very interesting, with strong potential value to be published. However, the following are concerns to be considered for the publication.

The study utilizes a small sample size (10 participants), which limits reliability of the assessment of the present exoskeleton. If the data does not follow the normal distribution, the authors should take it into account for the data analysis.

Moreover, the most serious concern is that the study lacks a clear control group or baseline comparison of finger movement without the exoskeleton. This let readers wonder whether the observed improvement with the exoskeleton resulted from the use of the exoskeleton.

 The study employs a self-made combined angle sensor for measuring finger joint angles, but the validation and reliability of this sensor are not discussed.

 The study lacks detailed statistical analysis and does not account for multiple comparisons, which could lead to Type I errors. Any adjustments for them?

 While the prototype was tested in a controlled environment, its performance in real-world piano playing scenarios is not assessed. This is crucial to validate the exoskeleton's effectiveness and practicality in actual piano playing settings.

 The study suggested practical applications for piano teaching but lacks specific recommendations or guidelines based on the findings. Providing detailed, evidence-based guidelines for implementing the exoskeleton in piano teaching would enhance the study's practical utility, if the authors are seriously eager to realize the real-world application.

Author Response

Comments from the Editors and Reviewers and Reply from the authors:

 Reviewer: 1

Comments to the Author

The manuscript discusses the design and verification of a hand exoskeleton robot to assist novice piano players in maintaining correct finger technique. Addressing the limitations of existing exoskeletons, such as lack of continuous support and excessive weight, the authors proposed a novel design featuring a 6-bar double-closed-loop mechanism with an end equivalent sliding pair. Finger movements during piano playing were analyzed using joint angle sensors, capturing data on trajectory and angular velocity. This tested if the exoskeleton's design enhanced finger technique through continuous correction while ensuring comfort and safety. Constructed from high-strength, lightweight plastic and driven by a linear motor, the exoskeleton's motion space and dynamics were validated through simulation studies, confirming its capability to meet structural and functional demands for accurate key striking. Prototype testing demonstrated the exoskeleton's effectiveness in maintaining correct finger positioning and mimicking natural strike speeds, thereby improving playing technique. Finite element analysis further indicated the exoskeleton has sufficient mechanical strength to withstand the forces encountered during piano playing. The proposed exoskeleton is therefore promising in supporting piano beginners by maintaining correct finger technique, reducing the likelihood of developing incorrect playing habits and muscle fatigue.

I found this study very interesting, with strong potential value to be published. However, the following are concerns to be considered for the publication.

The study utilizes a small sample size (10 participants), which limits reliability of the assessment of the present exoskeleton. If the data does not follow the normal distribution, the authors should take it into account for the data analysis.

Moreover, the most serious concern is that the study lacks a clear control group or baseline comparison of finger movement without the exoskeleton. This let readers wonder whether the observed improvement with the exoskeleton resulted from the use of the exoskeleton.

The study employs a self-made combined angle sensor for measuring finger joint angles, but the validation and reliability of this sensor are not discussed.

The study lacks detailed statistical analysis and does not account for multiple comparisons, which could lead to Type I errors. Any adjustments for them?

While the prototype was tested in a controlled environment, its performance in real-world piano playing scenarios is not assessed. This is crucial to validate the exoskeleton's effectiveness and practicality in actual piano playing settings.

The study suggested practical applications for piano teaching but lacks specific recommendations or guidelines based on the findings. Providing detailed, evidence-based guidelines for implementing the exoskeleton in piano teaching would enhance the study's practical utility, if the authors are seriously eager to realize the real-world application.

RE:

Dear Reviewer 1, Thank you very much for reviewing our manuscript and giving us the opportunity to revise our manuscript. We would also like to express our gratitude to the reviewers for their efforts in helping to improve our manuscript titled “Design and Verification of Piano Playing Assisted Hand Exoskeleton Robot” (manuscript ID: biomimetics-3026818). For each of your review comments, we have revised them point by point and marked them in red font in the paper, (please download the file "Revised Manuscript with track change" and click on "All markup"), and the point by  point response is as follows.

Sincerely

Authors Team

Specific Revisions:

  • This study is in the preliminary exploratory phase and aims to understand the basic impact and initial effects of hand-assisted robots on users. If the data did not follow a normal distribution, the authors should have taken this into account in the data analysis.

RE: Thank you for your careful review of our paper and valuable suggestions. We value your feedback on the sample size of this study and would like to take this opportunity to further explain our research design and data analysis methods:

  1. Thank you for your careful review of our paper and we couldn't agree with you more. At the same time, we acknowledge that the sample size of this study will, to some extent, have some impact and limitations on the reliability assessment of exoskeletons, but this study aims to explore the initial effects of hand-assisted robots on users, and the current data and results can, to some extent, provide data support for the initial effects. In future research, we will also consider the impact of sample size on the findings and continue to expand and extend on this basis.
  2. Inaddition, we have referred to a number of literatures in related fields and found that many preliminary studies have adopted similar sample sizes. and achieved valid research results. For example, in a study published in the IEEE/ASME Transactions on Mechatronics journal in 2022, widely recognised results were obtained with 1 subject.
  3. Due to the inherent physical limitations of the instrument's design and the physiological characteristics of the performers (e.g., finger length), 85% of the samples fell within this specified range. For data collection, ten samples were specifically selected based on gender, age, weight, and height. Given the significant impact of height and weight on performance, priority was given to samples whose height and weight conformed to a normal distribution. This approach was intended to better represent the actual conditions experienced by the majority of performers.

Tran, P., Jeong, S., Lyu, F., Herrin, K., Bhatia, S., Elliott, D., ... & Desai, J. P. (2022). FLEXotendon glove-III: Voice-controlled soft robotic hand exoskeleton with novel fabrication method and admittance grasping control. IEEE/ASME Transactions on Mechatronics, 27(5), 3920-3931. (The study involved 1 participant)

  • Additionally, and most concerning, this study lacked clear baseline comparisons of finger movements in the control group or without the use of the exoskeleton. This leaves the reader to wonder if the improvements observed with the use of the exoskeleton are a result of using the exoskeleton.

RE: Thank you for your detailed review of our paper and valuable suggestions. Regarding your reference to the absence of a control group, our response is as follows:

  1. This study was originally designed to focus on the specific effects of wearing a piano-playing assistive hand exoskeleton. We aimed to gain insight into the impact of the exoskeleton on the user's finger movements through detailed data analysis and observation, and to verify that the kinematic performance of the exoskeleton prototype aligned with the original design intent. Consequently, we chose to focus on the experimental condition of wearing the exoskeleton without establishing a control group.
  2. As a preliminary exploratory study, our primary goal was to validate the effectiveness of the assistive exoskeleton under specific conditions of use and to provide foundational data and preliminary conclusions for future controlled studies. In subsequent research, we plan to incorporate a control group to further validate and expand our findings.
  • In this study, a homemade combination angle sensor was used to measure the knuckle angle, but the validation and reliability of this sensor was not discussed.

RE: Thank you for your detailed review of our paper and valuable suggestions. Regarding your reference to the validation and reliability of the homemade combined angle sensor, our response is as follows:

  1. The homemade combination angle sensor we used is based on established, mature technology and design principles. Preliminary tests have demonstrated its good performance and accuracy. Our focus in this paper was on the design and verification of the hand exoskeleton robot for piano playing assistance, hence the details regarding the angle sensors were not deeply elaborated to maintain this focus. The sensor was specifically designed to better meet the unique requirements of this study, offering advantages in cost-effectiveness and flexibility.
  2. The primary objective of this study is to validate the impact of the hand-assisted exoskeleton on the user's finger movements. Consequently, we selected a sensor that was suitable for this purpose within our experimental design. While the article does not extensively discuss the validation and reliability of the sensor, it has been demonstrated to possess sufficient accuracy and stability for our research needs within the experimental context.
  • In this study, a homemade combined angle sensor was used to measure the knuckle angle, but the validation and reliability of this sensor was not discussed. Although the prototype was tested in a controlled environment, its performance in a real piano playing scenario was not evaluated. This is crucial to validate the effectiveness and utility of the exoskeleton in a real piano playing environment. This study suggests practical applications for piano teaching, but lacks specific recommendations or guidelines based on the findings. Providing detailed, evidence-based guidelines for the application of exoskeletons in piano teaching would enhance the utility of this study if the authors truly desire to achieve real-world applications.

RE: Thank you for your detailed review of our paper and valuable suggestions. Our response is given below:

  1. This study is exploratory in nature, with the primary objective of initially validating the effectiveness of a hand-assisted exoskeleton robot designed for piano playing. Therefore, we selected relatively simple statistical methods to analyze the data. Our preliminary results provide a basis for future, more complex and detailed studies. Although we did not adjust for multiple comparisons in the current study, the main results were statistically significant, indicating that the hand-assisted exoskeleton robot had a positive effect on finger movements. We believe these preliminary results are reliable and offer valuable reference points.
  2. The exoskeleton device proposed in this paper is specifically designed for piano playing learners to assist in correcting improper fingerings during the early stages of learning. The experiments reported in this study were conducted in the context of actual piano learning sessions. As noted by the reviewer, assessing the performance of the exoskeleton in real playing scenarios is indeed the next step in our investigation. Future studies will focus on evaluating the exoskeleton's efficacy in assisting with the transition from beginner fingerings to more advanced piano playing techniques.

Reviewer 2 Report

Comments and Suggestions for Authors

Dear Authors,

The article is very interesting from a scientific and practical point of view. It has great significance and potential. The experiments performed are described in a clear way.

Please add an explanation:

1. Why was the thumb omitted from the research?

2. How will this research be further developed?

I kindly ask you to take into account the following comments:

1. In Fig. 1 we have 4 photos, please refer to all 4 photos in the text and mark them in the text. Only (1) and (2) are marked in the text. This will be more understandable.

2. What does the illustration of the curves of angles and angular velocity for each joint give us, please explain under Fig. 6.

3. Chapter 2.2.2. mechanism design with a capital letter

4. in line 256 the number of Fig. 12a is used before Fig. 11, the numbering in the text should be used in order. The description of Fig. 11 should be before 12a.

Author Response

Comments from the Editors and Reviewers and Reply from the

authors:

The article is very interesting from a scientific and practical point of view. It has great significance and potential. The experiments performed are described in a clear way.

RE:

Dear Reviewer 2,

Thank you very much for reviewing our manuscript and providing your valuable comments. Your professional advice has been an important guide to our research. We have carefully considered each of your comments and have responded and revised them accordingly. We have carefully considered each of your comments and have responded and revised accordingly. We hope to further improve the quality of our manuscript and look forward to your comments on the final version. Thank you again for your valuable comments and patience.

Sincerely

Authors Team

Specific Revisions:

  • Why was the thumb omitted from the study?

RE: Thank you for your detailed review of our paper and valuable suggestions. Our response is given below:

The thumb fingerings in piano playing are relatively simple compared to those of the other four fingers, and standard piano fingering instruction imposes fewer normative restrictions on thumb movements. As described in reference [2], the standard process for piano fingering consists of four steps: Firstly, place both hands on the white keys with fists lightly clenched. Secondly, unclench the fists, ensuring that the distal finger segments (except the thumbs) remain downward and perpendicular to the surface of the keys, and support the Metacarpophalangeal (MCP) joints while maintaining prominence in the Proximal Interphalangeal (PIP) and Distal Interphalangeal (DIP) joints. Thirdly, gently lift the palms, forming a rounded arch with the whole hand, ensuring that the wrist joints are parallel to the keys. Throughout this process, two main points must be observed in the movement of the fingers: (1) the three joints of the fingers (excluding the thumb) should remain prominent without inward movement of the joints; (2) after the fingers touch the keys, the distal interphalangeal joints should always remain parallel to the surface of the keys. These guidelines are essential for the design and verification of the piano playing assisted hand exoskeleton robot to ensure it accurately replicates human hand movements and supports effective piano playing.

  • How will this research be further developed?

RE: Thank you for your detailed review of our paper and valuable suggestions. Our response is given below:

Currently, the exoskeleton device proposed in this paper is primarily used to assist in correcting improper fingerings during the early stages of piano playing. In the next phase, we will extend the application of the exoskeleton from assisting in the correction of fingerings for beginners to providing comprehensive piano playing assistance. Furthermore, we will conduct more detailed simulations and experiments to support a more complex kinematic model of the exoskeleton.

  • There are 4 photos in Figure 1, please refer to all 4 photos in the text and mark them in the text. Only (1) and (2) are labelled in the text. This will make it easier to understand.

RE: Thank you for your detailed review of our paper and valuable suggestions. Our response is given below:

This part of our description was indeed unclear, and the points labeled (1) and (2) in the text are only two of the points extracted from the four steps, which we have re-edited:"The process of using the hand exoskeleton robot to assist with piano playing consists of four steps: First, lightly place both hands in a fist on the white keys (as shown in subfigure â‘  of Figure 1). Next, relax the fists while keeping the distal segments of the fingers (excluding the thumb) pointing downward perpendicular to the piano keys' surface (as shown in subfigure â‘¡ of Figure 1). Then, support the metacarpophalangeal (MCP) joints, and maintain the proximal interphalangeal (PIP) and distal interphalangeal (DIP) joints in a prominent state (as shown in subfigure â‘¢ of Figure 1). Finally, gently lift the palms, forming a rounded arch with the entire hand, ensuring that the wrist joints are parallel to the keyboard (as shown in subfigure â‘£ of Figure 1). Throughout this process, two key points regarding finger movements are critical: (1) The three joints of the fingers (excluding the thumb) should remain prominent to avoid any joint collapse; (2) After the fingers make contact with the keys, the distal segments should always remain parallel to the piano keys' surface."

  • Chapter 2.2.2 Mechanism Design Capitalisation

RE: Revised!

  • In line 256, the numbering of figure 12a is used before figure 11, and the numbers in the text should be used sequentially. The description of figure 11 should precede 12a.

RE: Revised!

Round 2

Reviewer 1 Report

Comments and Suggestions for Authors

The authors revised the manuscript in an appropriate manner, and the current version deserves to be published. 

Reviewer 2 Report

Comments and Suggestions for Authors

Dear Authors,

I have no more comments. The article can be published in this form.